


# Rising Arctic Seas and Thawing Permafrost: Uncovering the Carbon Cycle Impact in a Thermokarst Lagoon System in the outer Mackenzie Delta, Canada

Maren Jenrich[1,2], Juliane Wolter[1,3], Susanne Liebner[3,4], Christian Knoblauch[5,6], Guido Grosse[1,2], Fiona Giebler[1,2], Dustin Whalen[7] and Jens Strauss[1]

[1]Alfred Wegener Institute Helmholtz Centre for Polar and Marine Research, Permafrost Research Section, Potsdam, 14473, Germany
[2]University of Potsdam, Institute of Geosciences, Potsdam, 14469, Germany
[3]University of Potsdam, Institute of Biochemistry and Biology, Potsdam, 14469, Germany
[4]GFZ German Research Centre for Geosciences, Section Geomicrobiology, Potsdam, 14473, Germany
[5]Universität Hamburg, Institute of Soil Sciences, Faculty of Mathematics, Informatics and Natural Sciences, Hamburg, 20148, Germany
[6]Universität Hamburg, Center for Earth System Research and Sustainability, Hamburg, 20148, Germany
[7]Geological Survey of Canada Atlantic, Natural Resources Canada, Dartmouth, Nova Scotia, B2Y 4A2, Canada

*Correspondence to*: Maren Jenrich (maren.jenrich@awi.de)

**Highlights**

- Highest total greenhouse gas production in the first stage of land-sea transition (active layer and young thermokarst lagoons under brackish conditions)
- $CO_2$ production increases along a land-sea transect with increasing salinity
- $CH_4$ production is highest under brackish conditions in lagoons
- Lowest overall production in first time thawed permafrost sediment, showing that microbes need time to adapt

**Abstract.** Climate warming in the Arctic is directly connected to rising sea levels and increasing erosion of permafrost coasts, leading to inland migrating coastlines and the transformation of coastal permafrost lakes into thermokarst lagoons. These lagoons represent transitional zones between terrestrial to subsea permafrost environments. So far, the effect of the transition on the carbon cycle is fairly unknown. In this study, we conducted long-term anoxic incubation experiments on surface samples from thermokarst lagoons with varying degrees of sea connectivity. We also included terrestrial permafrost and active layer as endmembers to investigate variations in carbon dioxide ($CO_2$) and methane ($CH_4$) production within lagoon systems and along a land-sea transition transect on Reindeer Island, Northeast Mackenzie Delta, Canada. Results show that $CH_4$ production peaks at 4.6 mg CH4 $gC^{-1}$ in younger, less connected lagoons with high-quality organic matter, leading to up to 18 times higher GHG production (in $CO_2$ equivalents) compared to open lagoons. $CO_2$ production is higher under marine conditions (3.8 to 5.4 $mgCO_2\ g^{-1}C$) than under brackish conditions (1.7 to 4.3 $mgCO_2\ g^{-1}C$). Along a land-sea transect, $CO_2$ production increased with increasing marine influence. These findings suggest that the landward migration of the sea, resulting in the inundation of permafrost lowlands and thermokarst lakes, may lead to increased GHG emissions from Arctic coasts in the future.





**Plain Language Summary.** Climate warming in the Arctic is causing the erosion of permafrost coasts and the transformation of permafrost lakes into lagoons. To understand how this affects greenhouse gas (GHG) emissions, we studied carbon dioxide ($CO_2$) and methane ($CH_4$) production in lagoons with varying sea connections. Younger lagoons produce more $CH_4$, while $CO_2$ increases in more marine conditions. Flooding of permafrost lowlands due to rising sea levels may lead to higher GHG emissions from Arctic coasts in the future.

## 1 Introduction


The Arctic region is experiencing unprecedented rates of temperature rise, nearly four times faster than the global average (Rantanen et al., 2022). The rapid warming has significant effects, particularly evident in the Arctic Ocean and its coastal areas, resulting in rising sea levels (Guimond et al., 2021; Nerem et al., 2018; Proshutinsky et al., 2001; Watson et al., 2015), extensive sea ice loss (Jahn et al., 2024; Kwok and Rothrock, 2009; Notz and Stroeve, 2016), and erosion of ice-rich permafrost

coasts (Günther et al., 2013; Jones et al., 2018; Malenfant et al., 2022; Whalen et al., 2022).

With approximately one million people residing along the Arctic coastlines, they are directly impacted by the transformation of coastal permafrost landscapes and the migration of the sea inland (Ramage et al., 2021). The average rate of permafrost coastline change throughout the Arctic between 1950 and 2000 was -0.5 m yr$^{-1}$, with significant regional and intraregional variations (Irrgang et al., 2022). The biggest rise in Arctic erosion rates, ranging from +80 to +160%, was observed along the

permafrost coasts of the Alaska and Canadian Beaufort Sea (Jones et al., 2020). In the Mackenzie Delta area, shoreline retreat rates have been documented as high as -46 m yr$^{-1}$, leading to substantial land loss and inland migration of the coastline (Malenfant et al., 2022; Solomon, 2005).

As sea levels rise and coastal erosion progresses, previously isolated thermokarst lakes and basins with bottom elevations below sea level become connected to the ocean, forming thermokarst lagoons (Jenrich et al., 2021; Schirrmeister et al., 2018).

Along the Arctic coast, from the Taimyr Peninsula in Russia to the Tuktoyaktuk Peninsula in Canada, 520 thermokarst lagoons larger than 500 metres in diameter have been identified, covering a total area of nearly 3500 km$^2$ (Jenrich et al., 2024b). These lagoons, located at the interface between land and sea, form a critical transition zone from terrestrial to subsea permafrost environments. Electrical resistivity surveys revealed that permafrost degradation in former lagoon deposits occurs up to 170% faster than in submerged Yedoma permafrost (Angelopoulos et al., 2021), significantly accelerating carbon cycling in these

areas.

Permafrost, which is ground that has been frozen for more than two consecutive years, lies beneath $13.9 \times 106$ km2 (ca. 15%) of the exposed land area (Obu, 2021) and about 407.680 km of arctic coasts (34% of the world coastline) are classified as permafrost coasts (Lantuit et al., 2012). Including also non-permafrost deposits the permafrost region stores ~1700 Pg organic carbon in active layer soils and deposits, frozen ground (Lindgren et al., 2018; Miner et al., 2022; Schuur et al., 2022), and

lake taliks (Anthony et al., 2014; Strauss et al., 2021), which is about 50% of the OC stored in all global soils (3350 Pg; Strauss



et al., 2024). Additionally, model simulations estimate another ~2800 Pg OC in deep permafrost below the seafloor of Arctic shelves (Miesner et al., 2023), which cover an area of almost $2.5 \times 10^6 \, \text{km}^2$ (Overduin et al., 2019). Rising temperatures in a warming Arctic accelerate permafrost thaw, activating ancient microbes and leading to the decomposition of organic carbon into greenhouse gases (GHGs) such as carbon dioxide ($CO_2$) and methane ($CH_4$).

The transition from unsaturated, aerobic conditions in terrestrial permafrost to saturated, anaerobic conditions in inundated soils results in a shift in microbial composition and, consequently, greenhouse gas production (Bush et al., 2017; Jenrich et al., 2024a; Liu et al., 2022). Under laboratory conditions, carbon breakdown processes and the associated production of $CO_2$ and $CH_4$ during landscape changes can be replicated by incubation experiments (e.g. Laurent et al., 2023; Liu et al., 2022; Tanski et al., 2019). $CO_2$ production is dominant in unsaturated soils, while $CH_4$ production is happening in saturated, anoxic soils

(Le Mer and Roger, 2001; Schädel et al., 2016). Marine sediments generally exhibit lower decomposition rates compared to terrestrial permafrost (Miesner et al., 2023). As terrestrial thermokarst lakes transition to marine environments, the hydrochemical and biogeochemical conditions of the sediment shift, affecting the microbial community composition (Jenrich et al., 2024a; Yang et al., 2023). In marine sediments, methanogens are found mostly in sediment layers below the sulfate reduction zone, as sulfate-reducing bacteria (SRB) outcompete methanogens for key substrates like acetate and hydrogen

(Jørgensen, 2006; Oremland and Polcin, 1982). During this transition, as soils become inundated with seawater, $CO_2$-producing SRB grow (An et al., 2023). A few studies focussing on organic-rich sediments (Holmer and Kristensen, 1994; Jørgensen and Parkes, 2010), salt marshes (Oremland et al., 1982), coastal sediments (Maltby et al., 2018) as well as in the early stages of lagoon formation (Jenrich et al., 2024a; Yang et al., 2023) have reported the coexistence of methanogens and SRB, which is probably driven by noncompetitive substrates like methanol and methylated compounds. A recent study simulating greenhouse

gas production during different transition stages in a coastal thermokarst landscape demonstrated that $CO_2$ production initially decreases following lagoon formation. However, in the long term, after SRB establish themselves, $CO_2$ production increases significantly, surpassing that in thawing terrestrial permafrost by a factor of eight (Jenrich et al., 2024a).

When highly degraded, lowland thermokarst landscapes are flooded by the sea, and thermokarst lagoons, lakes, and the sea become connected with each other, complex lagoon systems are formed. Consequently, a natural gradient of marine

submergence age and connectivity emerges within the system with older, well-connected lagoons closer to the sea, and younger, less connected lagoons further inland. The individual basins of previous thermokarst lakes are often still identifiable by their characteristic round shape in the new lagoon system. These systems are widespread along all Arctic lowland coasts with ice-rich permafrost and abundant thermokarst lakes, such as the Laptev, East Siberian, Chukchi, and Beaufort seas. They serve as a natural study setting for investigating the impact of progressive marine inundation on permafrost and its organic

matter pool. Different lagoon development stages found in the same region would represent a space-for-time gradient that could allow better understanding of the production of greenhouse gases in inundated permafrost soils under varying degrees of seawater influence. In Jenrich et al. (2024b), thermokarst lagoons are classified by openness and the degree of connectivity with the sea, both for single lagoons and lagoons within a lagoon system. They defined five connectivity classes from very high (5) for lagoons which are always open and in direct exchange with the sea to very low (1) for nearly-closed lagoons where





the exchange is very limited due to long channels or for subsequent lagoons in lagoon systems with very limited exchange due to narrow inlets and great distances to the sea.

Despite the abundance of thermokarst lagoons and their potential interesting role in the Arctic carbon cycle, their GHG production dynamics during the transition from terrestrial permafrost to subsea permafrost remain poorly studied so far. In particular, knowledge gaps persist regarding the variation in carbon degradability and GHG production based on lagoon

development stage as well as the evolution of GHG production with increasing seawater influence.

In this study, we address these gaps with answering the following questions:

1. How does greenhouse gas production differ within a lagoon system under near-natural conditions, considering different degrees of lagoon openness and distance to the sea?

2. How does greenhouse gas production differ with increasing seawater influence in the land-sea transition along a permafrost-

lake-lagoon-transect?

This study is the first focussing on thermokarst lagoon development stages and the GHG production rates occurring during the transition of permafrost from a terrestrial to a marine setting to better understand the role of thermokarst lagoons in the Arctic carbon cycle and more broadly what happens when terrestrial permafrost becomes inundated by the sea.

**2 Study Area**

In this study, we collected and incubated surface samples of thermokarst lagoons of different transgression stages within a lagoon system as well as terrestrial samples from the active layer and permafrost on Reindeer Island, northeast Mackenzie Delta, Canada.





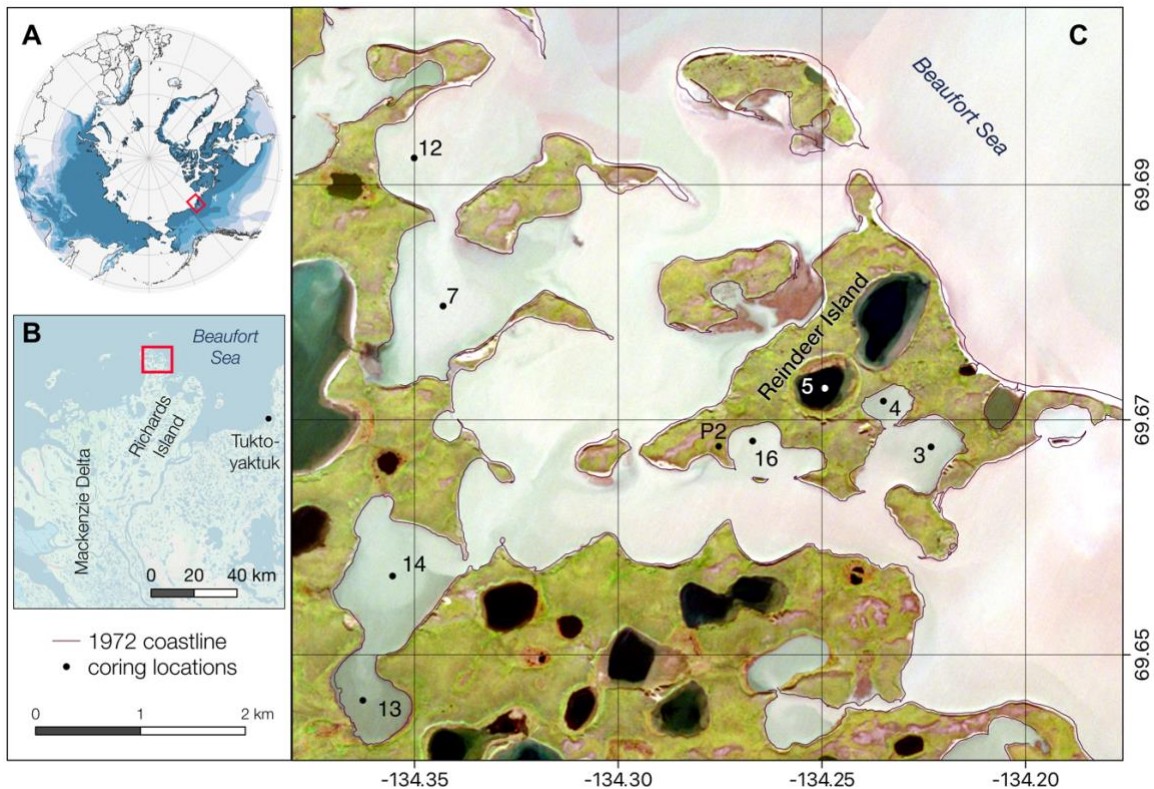

**Figure 1:  Study sites located on Reindeer Island and the surrounding lagoon system close to Richards Island, northern Mackenzie**
**Delta, Canada (A, B). Coring sites are marked by a dot C): P2 soil pit - active layer and permafrost sampling; 5 - thermokarst lake;**
**3, 4, 7, 12-14, and 16 - thermokarst lagoons of varying connectivity. Imagery sources: A) Permafrost extent regions based on Brown**
**et al. (1997); B) ESRI base map; C) Sentinel-2 satellite image band combination 4-3-2 from 2021-08-26. Note: different water colours**
**are related to sediment load coming from the Mackenzie River plume and indicative of the different lagoon connectivity, with black**
**colours being closed lake basins, darker grey colours limited-open or well-connected lagoon, and light grey waters in very open**
**lagoons (similar the open sea)**

Reindeer Island, formerly connected to the northern head of Richards Island, is situated in the northern region of the Mackenzie

Delta, Northwest Territories, Canada (Fig. 1). Richards Island is heavily influenced by thermokarst lake formation, covering

23.5% of its surface area (Burn, 2002). Seismic profiles revealed thaw depressions over 20 metres deep, half-filled with

stratified sediments, while radiocarbon and $^{137}$Cs data suggested sedimentation rates of 0.03±0.05 cm yr$^{-1}$, indicating lake

formation over the past 10,000-15,000 years (Solomon et al., 2000). The permafrost on Richards Island is continuous,

exceeding 500 m in thickness, contrasting sharply with the discontinuous permafrost of the Mackenzie Delta (Judge et al.,

1987; Kohnert et al., 2017). The regional sedimentology is characterised by discontinuous unconsolidated till in an ice-rich

permafrost environment. During the short open water (ice-free) season the area is exposed to high winds and increased wave

action during coastal storms. Although the southside of Reindeer Island and most of the inner lagoon is protected from the

forces of offshore waves during storms,  it is still susceptible to thawing permafrost during increased air temperatures. There



is a direct correlation to NW winds (>50 km/hr) and air temperature to coastal erosion in this region (Berry et al., 2021; Lim et al., 2020). Richard Island has experienced moderate to rapid coastal erosion rates, Hynes et al. (2014) reported erosion rates for 1972-2000 to be up to 3.6 m yr$^{-1}$, with significant annual retreat occurring at North Head (the tip of the Richards Island) and the ocean side of Reindeer Island. The average erosion rate within the lagoon study area was reported to be 0.28 m yr$^{-1}$
(Hynes et al., 2014), with increased rates (up to 1.1 m yr$^{-1}$) at some locations on the southside of Reindeer Island near samples sites 3, 4 and 16. Despite the regional and localised erosion it is clear that a significant portion of sediment are originating from the Mackenzie River's sediment plume (Solomon et al., 2000), which is supplying 128 Mt fluvial sediments on average per year (from 1974 to 1994; Carson et al., 1998). The Mackenzie River therefore is the largest supplier of fluvial sediments
and the fourth-largest provider of freshwater to the Arctic Ocean (330 km$^2$ yr$^{-1}$) (Vonk et al., 2015). 55% of the sedimentary OC is transported through the delta and deposited on the coastal shelf, suggesting that 45% are deposited mainly in lakes within the delta in the course of the spring ice break (Vonk et al., 2015) and when lakes get more connected to the river system (Burn, 1995) but likely also in the thermokarst lagoons at the head of the delta.

To the east, Reindeer Island borders the Beaufort Sea, while to the south and west, flooded thermokarst lake basins form a lagoon system separating the island from Richards Island. This lagoon system, consisting of at least 14 individual former thermokarst lake basins, covers an area of 13.7 km$^2$ (Jenrich et al., 2024b) and was first investigated in 1993 (Solomon et al., 2000). Bathymetric investigations during these studies have revealed overall shallow water depths (< 2 m) around Reindeer Island, allowing for the formation of bottom fast ice during winter. $^{137}$Cs data suggest that sedimentation rates in the lagoons
ranged from 0.7 to 2 cm yr$^{-1}$ of the past 50 years (Solomon et al., 2000). For this study we revisited five of the coring sites investigated by Solomon et al. (2000) (our site site names: LAG 3, 4, 7, 12 and TKL 5) and added three more (LAG 13, 14, 16), all representing lagoons with varying connectivity with the sea and therefore varying influence of marine processes and fluvial inputs from the Mackenzie River. Based on the classification by Jenrich et al. (2024b), LAG7 and LAG12 are Class 5 lagoons with very high connectivity, which are always open. LAG3 and LAG16 are considered Class 4, which are defined as
mostly open, spatially second-tier, lagoons which are very well connected to the primary lagoon. LAG4 and LAG14 are Class 3, semi-open second-tier lagoons which are well connected to the primary lagoon. LAG13 is a Class 2 second-tier lagoon, which is less connected to the primary lagoon due to the great distance to the sea and multiple narrow inlets reduce the exchange with the sea.

Further we sampled active layer and permafrost in a low land position (P2) on the island.

## 3 Methods

### 3.1 Fieldwork and Subsampling

Fieldwork was conducted at the Reindeer Island lagoon system in collaboration with Natural Resources Canada during August 2021. For this study we chose 7 distinct locations within the lagoon system as well as one thermokarst lake. Using a boat we





took surface water samples using a water sampler (UWITEC, Austria), collected Conductivity-Temperature-Depth (CTD)

information by using a CastAway CTD device (SonTek, USA), and cored sediments using a gravity coring system (UWITEC Single Corer, Austria). A minimum of 2 cores were extracted from each site, with one core designated for microbiological analyses and subsequent incubation experiments. These cores were preserved within gas-proof bags filled with nitrogen gas to maintain anaerobic conditions and stored at a temperature of 4°C. A second core was reserved for sedimentological analyses and also stored at 4°C. The length of the cores varied from 10 cm to 49 cm.

Furthermore, active layer (AL) and permafrost samples were taken from a soil pit located on the lowland of Reindeer Island. The stratigraphy of the active layer was described and documented, with subsamples being selectively collected based on visual observations of distinct sediment layers. These subsamples were used for incubation experiments, including microbiological analyses and sediment analyses. Dimensions of the subsamples were recorded to calculate sample volumes for subsequent bulk density calculations. In addition, permafrost sampling was conducted using a Hilty hand drill, extracting

uppermost permafrost samples extending to a depth of 12 cm. These permafrost samples were kept frozen in the field and transported to the Alfred Wegener Institute laboratories in Potsdam for further analysis.

Samples for incubation experiments were placed in precombusted glass jars, sealed and stored within nitrogen-filled gas proof bags. Biomarker samples were preserved in precombusted sterilised glass jars, and sediment samples for subsequent analyses were stored in sterile plastic bags (WhirlPack).

In the laboratory, the sediment cores were cut length-wise and the visual stratigraphy was described and documented. Subsamples were carefully extracted ranging from depths of 3 to 10 cm, intended for incubation and microbiological analyses. The edges of these subsamples were cleaned by scratching off the outermost material, and the sample material was deposited into pre-combusted glass jars. To minimise contact with oxygen, these jars were immediately flushed with nitrogen gas and subsequently stored in bags filled with nitrogen at a temperature of 4°C until the initiation of the planned experiments.

Moreover, subsamples from one core half were taken every 5 cm for sedimentology and geochemistry analyses, while the other half was preserved in an archive freezer for future reference. The wet weight of all subsamples was recorded before further handling.

### 3.2 Hydrochemistry

Using Rhizon samplers (membrane pore size: 0.12-0.18 μm), pore water was collected from thawed samples for the hydrochemical pre-incubation analysis. The pH and electrical conductivity (EC in mS/cm) of the pore water were measured using a WTW Multilab 540, with an accuracy of ±0.01 for pH and ±1 mV for EC. After treating the DOC samples with 50 μL of 30% HCl supra-pure, they were stored at +4°C until analysis with a Shimadzu Total Organic Carbon Analyser (TOC-VCPH) following the protocol described by Fritz et al. (2015), with an accuracy of ±1.5%. Pore water for measuring sulfate

concentration was diluted (1:50) and subjected to triplicate analysis using the Sykam S155 Compact IC-System ion



chromatograph, which has a detection limit of 0.1 mg/L. The detected peaks in the chromatograms were automatically integrated using the ChromStar 7 program, and the triplicate average was used for further assessment.

Using the TEOS-10 MATLAB implementation, we were able to convert the measured electrical conductivity (with respect to 25 °C) to molality (mol/kg) and absolute salinity (g/kg) (McDougall & Barker, 2011). This conversion package assumes that

the pore water fluid is consistent with standard seawater composition (Millero et al., 2008).

In order to be able to test GHG production in different sediments during the phases of landscape development (lake, lagoon, subsea), it is crucial to keep the boundary conditions (fresh: $c = 0$ g/L, brackish: $c = 13$ g/L, marine: $c = 36$ g/L) and the total water volume of 10.5 mL constant. For this purpose, we calculated based on the molarity of the pore water how much of the highly concentrated artificial seawater solution ($c = 182.55$ g/L) needed to be added to the samples. The artificial seawater

solution had a higher concentration than that of standard seawater, so a relatively lower volume of water could be added to the sediment pore water and be diluted. The artificial seawater contained of NaCl (24.99 g/L), $MgCl_2 \times 6H_2O$ (11.13 g/L), $Na_2SO_4$ (4.14 g/L), $CaCl_2 \times 2H_2O$ (1.58 g/L), KCl (0.79 g/L) and $NaHCO_3$ (0.17 g/L) dissolved in ultrapure water and sterile filtered after.

### 3.3 Sedimentological and biogeochemical bulk analyses

Prior to and following freeze-drying (using a Zirbus Sublimator 15), the sediment was weighed, and the weight difference between the wet and dry sediment was used to calculate the absolute water content.

Grain-size analyses were performed using a Malvern Mastersizer 3000 with a connected Malvern Hydro LV wet-sample dispersion machine on organic-free (processed with 35 percent $H_2O_2$) samples. The percentages of silt, clay, and sand are given as sums between 2 mm and 63 μm, 63 μm and 2 μm, and less than 2 μm, in that order. Grain-size parameters were computed

using Gradistat (Blott and Pye, 2001; Version 8.0).

The total carbon (TC) and total organic carbon (TOC) content (expressed in weight percent, wt%) of homogenised and milled bulk samples (using a Fritsch Pulverisette 5 planetary mill) were analysed using a soliTOC cube, and the total nitrogen (TN) content was determined using a rapid max N exceed (both Elementar Analysensysteme, Langenselbold, Germany); both instruments had a device-specific accuracy of ± 0.1 wt% and a detection limit of 0.1 wt%. Carbonates were removed from

sediments using 1.3 molar hydrochloric acid (HCl) at 50 °C for five hours in order to perform stable carbon isotope analysis of organic carbon. The samples were then dried once more after being cleaned of chloride ions.

Next, at the AWI ISOLAB Facility Potsdam, stable carbon was analysed with a ThermoFisher Scientific Delta-V-Advantage gas mass spectrometer fitted with a CONFLO IV and a FLASH 2000 elemental analyser.

### 3.4 Incubation Experiment

To analyse how $CO_2$ and $CH_4$ production differs within a lagoon system (objective 1), given that well-connected lagoons (LAG7, LAG12, LAG14, LAG16) have been under seawater influence for a longer period and to a greater extent than poorly connected, younger lagoons (LAG4, LAG13), we incubated subsurface samples (3-10cm depth) of the lagoons anoxic under



brackish (c=13 g/L) and marine (c=36 g/L) conditions using artificial seawater for 415 days. As oxygen has a negative impact on anaerobic microbial populations, we disposed of the first three cm of the sample and kept the environment oxygen-free for the duration of the incubation and sample processing (handling and overnight thawing at 8 °C in a glovebox under $N_2$ atmosphere).

To investigate the variation in $CO_2$ and $CH_4$ production with increasing seawater influence in a land-sea transect (objective 2) we incubated permafrost, lake, and lagoon samples, 4, 3, 16) under increasing saline conditions as described in (Jenrich et al., 2024a). In short, we incubated the terrestrial samples (P2-P, P2-AL, TKL5) with freshwater to simulate near-natural lake conditions, brackish to simulate freshly formed lagoons, and with marine water to simulate established lagoons. We used sediments from the three lagoons (LAG4, LAG3, LAG16) because they differ in age and connectivity with the sea and therefore represent different states of lagoon development. The lagoon samples were incubated under brackish and marine conditions to simulate near-natural conditions depending on their pore water salinity. Permafrost and active layer samples have been incubated for 244 days while lagoon samples have been incubated for 415 days. Accordingly, the cumulative $CO_2$ and $CH_4$ after 244 days is used to answer question 2 (GHG production along land-sea transect).

In order to ensure similar boundary conditions for the samples and their individual treatments, we used the same dry weight (5g) and the same water content (10.5 mL) in all bottles, as well as the same salinity for the brackish and marine treatment. Therefore, depending on the initial pore water content and salinity, different amounts of wet soil had to be weighed in and different volumes of artificial seawater added for the samples respectively to the treatment (fresh: c = 0 g/L, brackish: c = 13 g/L, 373 marine: c = 36 g/L). We acknowledge that submerged sediments close to the coast of the Mackenzie Delta still experience brackish conditions, but for simplicity, we ignore the effects of river discharge.

Given the well-established anaerobic conditions in waterlogged soils in situ, we conducted the incubation anaerobically.

The samples were homogenised before being put into 120 ml glass incubation vials that had been previously combusted. For every treatment, three replicates were incubated at 4 °C for comparability with previous permafrost incubation studies (e.g., Jongejans et al., 2021; Tanski et al., 2019) and to roughly match with the temperature of the water at the bottom of the water bodies in the summer.

$CO_2$ and $CH_4$ concentrations were measured using gas chromatography (7890A Agilent, United States) equipped with a thermal conductivity detector and a flame ionisation detector, respectively, with helium as the carrier gas and a 100 °C oven furnace temperature. To prevent zonation in sediment and water, the incubation vials were shaken before each measurement. Using a gastight syringe, gas samples were extracted from the vials' headspace and promptly fed into the gas chromatograph. The measurements were taken five times in the first two weeks, then every week for the next seven weeks, and then roughly every two weeks after that. Gas concentrations were normalised to the dry weight of the sediment (gdw$^{-1}$). The total amount of $CO_2$ and $CH_4$ was calculated in µmol using the ideal gas law (Knoblauch et al., 2018) employing the gas concentration, headspace volume, water volume, pH, temperature, and solubility, including carbonate and bicarbonate concentrations for $CO_2$





calculations (Millero et al., 2007). Further, the average of $CO_2$ and $CH_4$ for each of the three replicates was calculated, and we used the TOC content to normalise the data to $gSOC^{-1}$. Rates were then determined using the measured time intervals. Cumulative values were obtained by subtracting the initial measurement from each subsequent one, allowing for the calculation
of production rates.

### 3.5 Statistics

All data analyses were conducted in R version 4.3.2 (R Core Team 2023). The Wilcoxon signed-rank test for paired samples was used to assess differences between brackish and marine treatments for all lagoons (n=7) and all transect sites (n=6), respectively. Principal component analyses (PCA) were performed on range transformed data using the base R function
princompto explore which biogeochemical parameters correlate with $CO_2$ and $CH_4$ production in the different connectivity classes. The parameters tested included surface water EC, TOC, TN, and $\delta^{13}C$ under both brackish and marine conditions. The PCA results were visualised in a biplot using the 'ggbiplot' package (Vu and Friendly, 2023).

### 4 Results

### 4.1 Surface water hydrochemistry

Table 1 summarises the surface water properties of the lake and the seven lagoons. Only two of the lagoons we examined (LAG3, LAG12) had a depth shallower than 2 m, while the others ranged between 2.3 m and 7.4 m in depth. Average EC over depth measured in the lagoons ranged from 5.8 to 17.5 mS/cm and is therefore much lower than the average surface water salinity of the Arctic Ocean (52.8 mS/cm, 33.8 psu; Hall et al., 2023).

During summer, stratification occurred in the deeper lagoons due to freshwater discharge from the Mackenzie River,
particularly evident in LAG7 with a temperature difference of approximately 8°C and an EC difference of around 27mS/cm between surface and near-bottom layers. No stratification was observed in the lake and shallow lagoons LAG12 and LAG13, while in LAG16, there was no difference in temperature but a variation in salinity.

**Table 1. Lake (TKL5) and Lagoon (LAG) surface water properties: Connectivity Class (2: low to 5: very high - open lagoons), water**
**depth, temperature (T), electrical conductivity (EC), and pH have been measured in the field in August 2021. Dissolved organic carbon (DOC) and sulfate concentrations were measured in the laboratory.  Min: minimal, max: maximal, av: avererage.**



| Site | Class | Latitude (decimal | Longitude degree) | Depth (m) | $T_{min}$ (°C) | $T_{max}$ (°C) | $T_{av}$ (°C) | $EC_{min}$ (mS/cm) | $EC_{max}$ (mS/cm) | $EC_{av}$ (mS/cm) | DOC (mg/L) | Sulfate (mg/L) | pH |
|---|---|---|---|---|---|---|---|---|---|---|---|---|---|
| TKL5 | | 69.6723 | -134.2493 | 12.9 | 10.0 | 10.9 | 10.4 | 0.3 | 0.4 | 0.3 | 3.7 | 2.9 | 8.5 |
| LAG13 | 2 | 69.6457 | -134.3627 | 2.3 | 9.7 | 9.7 | 9.7 | 10.6 | 11.5 | 11.1 | 5.0 | 671.0 | 8.2 |
| LAG4 | 3 | 69.6714 | -134.2341 | 4.6 | 4.6 | 8.3 | 6.6 | 11.6 | 17.5 | 14.0 | 5.5 | 656.0 | 7.9 |
| LAG14 | 3 | 69.6563 | -134.3555 | 6.4 | 6.4 | 9.8 | 8.9 | 10.3 | 16.3 | 12.9 | 5.0 | 660.0 | 8.2 |
| LAG3 | 4 | 69.6673 | -134.2214 | 1.1 | 5.6 | 8.6 | 7.1 | 10.5 | 17.8 | 13.2 | 4.9 | NA | 8.2 |
| LAG16 | 4 | 69.6678 | -134.2669 | 2.9 | 8.8 | 9.8 | 9.4 | 5.2 | 15.1 | 8.3 | 5.3 | 328.0 | 8.2 |
| LAG7 | 5 | 69.6792 | -134.3428 | 7.4 | 0.8 | 8.7 | 6.4 | 5.9 | 33.1 | 17.5 | 6.0 | 385.0 | 7.8 |
| LAG12 | 5 | 69.6918 | -134.3502 | 1.2 | 9.2 | 9.4 | 9.3 | 5.6 | 6.0 | 5.8 | 5.4 | 351.0 | 8.2 |

## 4.2 Bulk sediment biogeochemistry and pore water hydrochemistry

Results on bulk sediment biogeochemistry and pore water hydrochemistry are presented in Table 2. The TOC content of the
sediment was lowest in LAG3 (1.45 wt%) and highest in the active layer (26.31 wt%). Nitrogen content ranged from 0.17 %
in LAG3 and LAG12 to 1.53 % in the active layer, followed by permafrost and LAG13. The C/N ratio was lowest in LAG3
(8.53) and highest in permafrost (30.91). The average C/N ratio in the lagoons was approximately 14±3. Additionally, the $\delta^{13}C$
composition was relatively homogeneous, ranging from -27.65 ‰ in the AL to -25.86 ‰ in LAG3.

Regarding pore water, water content was lowest in the permafrost sample and highest in the AL. EC of the pore water varied
significantly, with the lowest salinity recorded in the freshwater lake (EC: 0.59 mS/cm, 0.02 g/L) and the highest in LAG7
(EC: 50.7 mS/cm, 33.4 g/L). DOC concentrations showed marked differences, with the lowest levels detected in the lake, LAG
3 and LAG 13 and the highest in the permafrost samples.

In surface water, sulfate concentration exhibited diverse patterns. The lowest levels were recorded in the lake (3 mg/L), while
the highest was observed in LAG13 (671 mg/L). Similar sulfate concentrations were detected for LAG7, 12, and 16, as well
as for LAG4, 13, and 14.

**Table 2. Bulk sediment and porewater biogeochemistry of permafrost (PF), active layer (AL), thermokarst lake (TKL05), and thermokarst lagoons (LAG): Total organic carbon (TOC) and nitrogen (N) content, carbon to nitrogen ratio, stable 13C isotopes, water content (WC), electrical conductivity (EC), salinity, pH, and dissolved organic carbon content (DOC).**



| Sample | Bulk sediment | | | | | Porewater | | | | |
|---|---|---|---|---|---|---|---|---|---|---|
| | sample depth [cm] | TOC [%] | N [%] | C/N | d13C [‰] vs. PDB | WC [%] | EC [mS/cm] | Salinity [g/L] | pH | DOC [mg/L] |
| P2_PF | 44.3 | 16.38 | 0.53 | 30.91 | -27.45 | 9.81 | 1.64 | 0.02 | 6.55 | 216 |
| P2_AL | 3.5 | 26.31 | 1.53 | 17.20 | -27.65 | 70.16 | 3.44 | 0.05 | 6.58 | NA |
| TKL05 | 5.5 | 2.72 | 0.25 | 10.88 | -27.15 | 52.74 | 0.59 | 0.29 | 7.05 | 5.03 |
| LAG13 | 5.5 | 6.38 | 0.46 | 13.87 | -26.64 | 56.84 | 20.75 | 12.46 | 7.08 | 7.08 |
| LAG04 | 5.5 | 2.84 | 0.23 | 12.35 | -25.99 | 57.43 | 48.10 | 31.49 | 7.17 | 20.14 |
| LAG14 | 5.5 | 2.95 | 0.21 | 14.05 | -26.34 | 46.82 | 29.06 | 18.02 | 7.27 | 18.06 |
| LAG03 | 4.5 | 1.45 | 0.17 | 8.53 | -25.86 | 27.84 | 16.44 | 9.67 | 7.32 | 7.76 |
| LAG16 | 5.5 | 2.99 | 0.18 | 16.61 | -26.34 | 40.92 | 36.60 | 23.23 | 7.16 | 26.65 |
| LAG07 | 5.5 | 2.98 | 0.18 | 16.56 | -26.45 | 48.23 | 50.70 | 33.40 | 7.40 | 16.95 |
| LAG12 | 5.5 | 2.65 | 0.17 | 15.59 | -26.35 | 33.59 | 33.40 | 21.00 | 7.40 | 23.90 |

### 4.3 Grain size distribution

The particle size distribution analysis shows significant differences between permafrost and the active layer (Fig. S1). Permafrost exhibited a predominance of fine silt particles, with a mean grain size of 8.1 µm, whereas the active layer was characterised by sandy sediment, with a mean grain size of 97.3 µm. Lagoons LAG4 to LAG16 displayed a consistent pattern of fine silt composition, with mean grain sizes ranging from 3.75 to 7.41 µm. However, Lagoon LAG3 stood out due to its very poorly sorted medium to coarse silt with a mean grain size of 34.2µm. LAG3 also has a second peak in the sandy fraction, which is very similar to the sandy active layer. It is also notable that the permafrost has a unimodal (one peak) distribution, as do most of the lagoons within a similar range as the permafrost, while the lake and LAG3 exhibit a bimodal (two peaks) distribution. The active layer is unimodal but has its peak in a different range. This suggests that at least two main processes contribute to the depositional regimes.



**4.4.1 $CO_2$ und $CH_4$ production in lagoon sediments after 415 days on incubation**

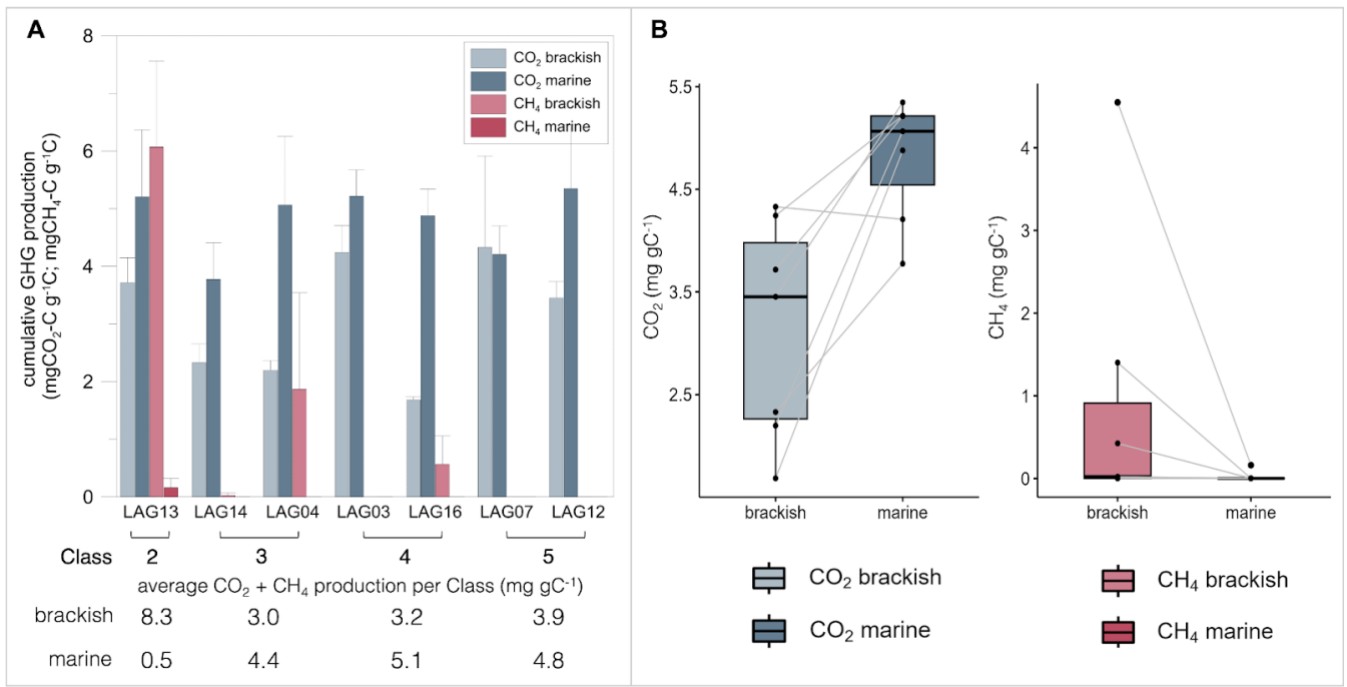

**Figure 2. Cumulative $CO_2$ (mg$CO_2$ g$^{-1}$C) and $CH_4$ (mg mg$CH_4$ g$^{-1}$C) production of surface sediment from the seven lagoons of varying seawater connectivity under brackish (13 g/L) and marine (36 g/L) conditions after 415 days of anaerobic incubation. A:**
**Grouped barplot with lagoons (LAG) ordered by connectivity class from low connectivity (Class 2) to very high connectivity (Class 5) to the sea. Error bars indicate standard deviation. B: Treatment comparison - paired box-whisker plots with connected data points (lines). The treatment has a significant effect on both $CO_2$ and $CH_4$ production (two sided Wilcoxon test: p=0.031 and p=0.047, respectively).**

The results on the cumulative anaerobic $CO_2$ and $CH_4$ production measured after 415 days for the lagoons are shown in Figure

335   2.

$CO_2$ production ranged from 1.68 to 4.33 mg$CO_2$ g$^{-1}$C under brackish and from 3.78 to 5.35 mg$CO_2$ g$^{-1}$C under marine conditions and the $CH_4$ production ranged from 0.00 to 4.55 mg$CH_4$ g$^{-1}$C under brackish and from 0.00 to 0.16 mg$CH_4$ g$^{-1}$C under marine conditions  (Figure 2A). $CO_2$ production was significantly lower under brackish, compared to marine conditions (p=0.016), while $CH_4$ production was significantly higher under brackish conditions (p=0.023) (Figure 2B). For six of the

seven lagoons no $CH_4$ production was observed under marine conditions. The highest cumulative $CH_4$ production rate was observed at brackish conditions below the limited-open LAG13 (4.55 ± 1.12 mg$CH_4$ g$^{-1}$C). $CH_4$ production started after 150 to 200 days for the LAG13, 4 and 16 under brackish conditions and for LAG13 after 360 days under marine conditions (Figure S1). For Lagoons 14, 3, 7 and 12 no methane production was observed in the timeframe of the experiment.




LAG14 and LAG04, both categorised with medium connectivity (Class 3), exhibit very similar $CO_2$ productions under brackish
conditions of $2.33 \pm 0.32$ mg$CO_2$ g$^{-1}$C and $2.20 \pm 0.17$ mg$CO_2$ g$^{-1}$C, respectively, while under marine conditions LAG14
produced more $CO_2$.

Mostly open Class 4 lagoons, LAG03 and LAG16, both categorised with high connectivity to the sea, demonstrate different
cumulative $CO_2$ productions under brackish conditions of $4.24 \pm 0.46$ mg$CO_2$ g$^{-1}$C and $1.68 \pm 0.05$ mg mg$CO_2$ g$^{-1}$C,
respectively but similar $CO_2$ production under marine conditions.

The very highly connected Class 5 lagoons, LAG07 and LAG12, both show high $CO_2$ production under brackish and marine
conditions. Highest $CO_2$ production of all lagoons was observed in LAG12 with $5.35 \pm 1.12$ mg$CO_2$ g$^{-1}$C.

**Relationships between biogeochemical parameters and gas production:** Based on the principal component analysis (Fig.
S2, S3) of the parameters brackish and marine $CO_2$/ $CH_4$ production, TN, TOC, $\delta^{13}$C, and surface water EC, the sites are
grouped by openness, with the most closed lagoon forming one group and the more open lagoons forming another. $CO_2$
production correlates with surface water EC but not clearly with substrate parameters, while $CH_4$ production correlates with
substrate parameters in brackish and marine incubations.





**4.4.1 CO₂ and CH₄ production of permafrost, lake and lagoon sediments after 244 days on incubation**

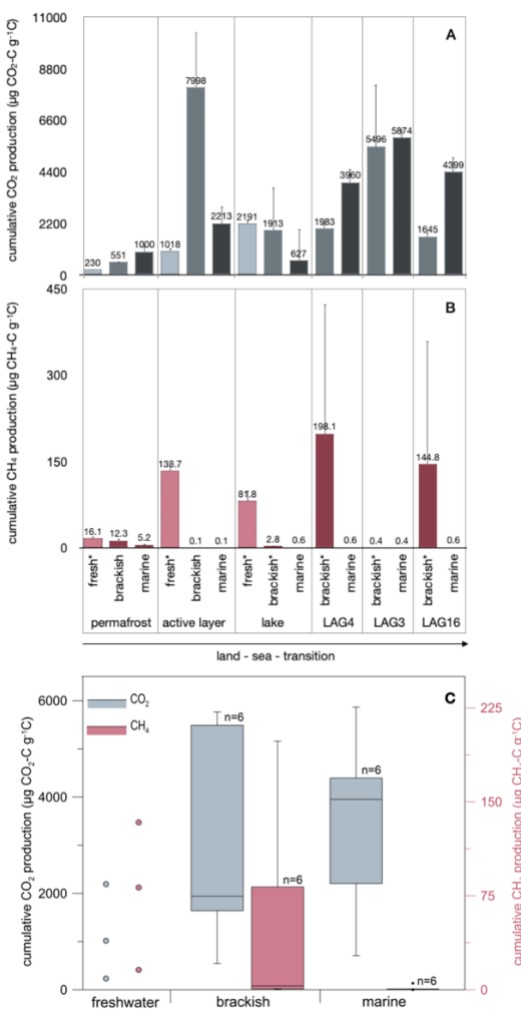

**Figure 3. Cumulative A) CO₂ and B) CH₄ production after 244 days of anaerobic incubation at 4°C under an increasing seawater influence (treatment: fresh, brackish, marine conditions) and C) Box-whisker plots visualising the effect of the treatment on CO₂ and CH₄ production. Note that CO₂ and CH₄ production are not at the same scale.**

The incubation with fresh water, which represents natural conditions for the terrestrial sites showed that the thermokarst lake sample exhibited the highest cumulative $CO_2$ production of $2191 \pm 136$ µgCO$_2$ g$^{-1}$C, while the permafrost sample showed the lowest $CO_2$ production at $230.49 \pm 16.95$ µgCO$_2$ g$^{-1}$C (Fig. 3A). The highest cumulative $CH_4$ production was measured in the active layer sample with $133.73 \pm 13.53$ µgCH$_4$ g$^{-1}$C (Fig. 3B). In contrast, the thermokarst lake sample exhibited the lowest $CH_4$ production at $81.78 \pm 17.14$ µgCH$_4$ g$^{-1}$C.

Under brackish conditions, the permafrost sample exhibited the lowest $CO_2$ production of $551.44 \pm 25.38$ µgCO$_2$ g$^{-1}$C. Even though brackish conditions are not natural for microbes in terrestrial settings, the active layer sample showed the highest





cumulative $CO_2$ production ($7997.90 \pm 2349.58$ µg$CO_2$ g$C^{-1}$), followed by the mostly open lagoon LAG3 ($5495.74 \pm 2631.84$ µg$CO_2$ g$^{-1}$C) (Fig. 3A). Regarding $CH_4$ production, the LAG4 sample displayed the highest cumulative production under brackish conditions with $198.10 \pm 224.93$ µg$CH_4$ g$^{-1}$C, while the active layer sample showed the lowest average $CH_4$ production at $0.2 \pm 0.15$ µg$CH_4$ g$^{-1}$C (Fig. 3B).

Under marine water treatments, the most $CO_2$ was produced in the LAG3 sample ($5874.11 \pm 313.20$ µg$CO_2$ g$^{-1}$C), while the lake sample showed the lowest $CO_2$ production ($626.79 \pm 1310.36$ µg$CO_2$ g$^{-1}$C) (Fig. 3A). Regarding $CH_4$ production, the permafrost sample displayed the highest cumulative production of $5.17 \pm 1.36$ µg$CH_4$ g$^{-1}$C, while the active layer sample showed the lowest $CH_4$ production at $0.07 \pm 0.03$ µg$CH_4$ g$^{-1}$C. However $CH_4$ production under marine conditions was generally low for all samples (Fig. 3B).

In Fig. 3C, $CO_2$ and $CH_4$ production is compared by treatment. By testing the effect of the treatment on $CO_2$ and $CH_4$ production in a paired Wilcoxon test for the 6 transect settings, we found that $CO_2$ production does not differ significantly between brackish and marine treatments ($p=0.844$), but $CH_4$ production is significantly higher under brackish than marine conditions ($p=0.031$). If all 10 sites are considered, the difference is stronger  (Wilcoxon test, paired, two-sided, $p=0.03711$).

**Relationships between biogeochemical parameters and gas production:** Based on the principal component analysis (Figures S4, S5) of the parameters brackish and marine $CO_2$/$CH_4$ production, TOC, $\delta^{13}$C, and surface water EC, terrestrial and lagoon sites group separately. $CO_2$ production in brackish incubations shows no correlations, while in marine incubations, $CO_2$ production is positively correlated with $\delta^{13}$C and surface water EC. $CH_4$ production in brackish incubations is positively correlated with $\delta^{13}$C and surface water EC, and negatively correlated with TOC, but shows no clear correlations in marine incubations.

**4 Discussion**

The Reindeer Island lagoon system was first studied in 1994 to investigate the response of the Mackenzie Delta shoreline to changing hydrological influences (Solomon et al., 2000). Using seismic data to estimate the volume of flooded lake basins and sediment fill on Richards Island, along with estimates of sediment input from the Mackenzie River plume, they show that most of the sediment in the marine areas comes from tidal exchange with inner shelf water and storm surges. The study estimated that Richards Island's thaw lakes contain 250,000 tonnes of total organic carbon (Solomon et al., 2000). Almost 30 years later we revisited the study area to investigate the greenhouse gas production within the lagoon system and along a land-sea transition transect to get a better understanding on carbon dynamics in these transitional environments.



## 4.1 Variations of $CO_2$ and $CH_4$ production within the lagoon system

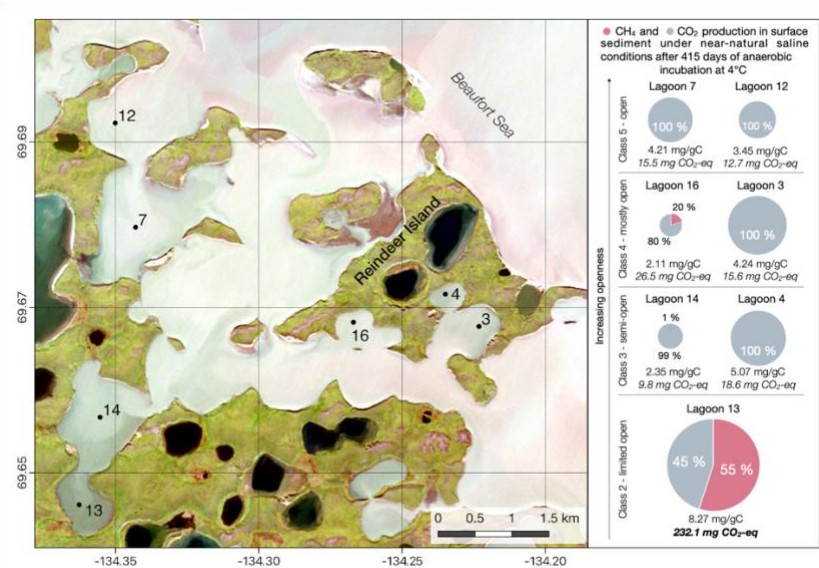

**Figure 4. Cumulative $CO_2$ and $CH_4$ production in subsurface lagoon sediments after 415 days of anaerobic incubation under near-natural salinity conditions which are brackish conditions for Lagoons 12, 16, 3, 14 and 13 and under marine conditions for Lagoons 7 and 4. The size of the pie diagram is proportional to the total production of $CO_2$ and $CH_4$ in mg g$^{-1}$C (number below pie). The share of $CH_4$ (pink) and $CO_2$ (grey) for the total production of $CO_2$ and $CH_4$ (in mg g$^{-1}$C) is given in percent. The total in $CO_2$ equivalents is based on 36 g$CO_2$eq/g$CH_4$ for GWP100 (Balcombe et al., 2018). Map: true colour sentinel 2 satellite image from 2021-08-26.**

We found that the production of greenhouse gases within a lagoon system changes with increasing connection to the sea. In Fig. 4, we show cumulative $CO_2$ and $CH_4$ production under near-natural saline conditions. Here, we define near-natural as brackish for pore water salinity of less than 24.5 g/L (mean of brackish (13 g/L) and marine (36 g/L) treatment), which was the case for LAG12, 16, 3, 14, and 13 and marine for pore water salinities greater than 24.5 g/L, LAG 7 and 4. In the youngest, less open LAG13 (Class 2), methane production was observed to be the highest (Fig. 2). Observing $CH_4$ production in sulfate-containing environments is surprising, as the current understanding is that sulfate-reducing bacteria (SRB) outcompete methanogens for the major substrates hydrogen and acetate (Holmes et al., 2017; Kristjansson and Schönheit, 1983; Lovley et al., 1982; Olefeldt et al., 2013; Schönheit et al., 1982). Nevertheless, $CH_4$ production under brackish conditions in young lagoons has been observed before (Jenrich et al., 2024a; Yang et al., 2023). In these environments, which are in the early stages of transitioning from a terrestrial lake to a lagoon, terrestrial methanogens are still present. Under low sulfate levels, methanogens and SRB can coexist (Dar et al., 2008; Yang et al., 2023). LAG13 shows the highest TOC concentrations, which additionally was less depleted in $^{13}$C. The higher availability of still high-quality organic matter for microbial decomposition is likely also driving this enhanced methane production. High $CH_4$ production is accompanied by high $CO_2$ production, indicating that in this initial stage of the land-sea transition, total GHG production reaches its peak. In terms of $CO_2$ equivalents, calculated based on 36 g$CO_2$eq/g$CH_4$ for a Global Warming Potential (GWP)100 (Balcombe et al., 2018), the $CO_2$-eq of



LAG13 is significantly higher—up to 18 times—compared to the open lagoons (e.g. LAG12). This is in line with the results of Jenrich et al. (2024a), where the highest $CO_2$ and $CH_4$ production was also detected for the most closed lagoon (Class 1 after Jenrich et al., 2024a) under brackish conditions. Consequently, young lagoons with high OM content and quality exert the greatest climate impact among the studied lagoons (Fig. S2). However, compared to current values, $CO_2$ production in the

lagoons of this study area is lower than that of the studied lagoons on Bykovsky Peninsula, Siberia(Jenrich et al., 2024a). There, the maximum $CO_2$ production reached up to 23 $mgCO_2$ $g^{-1}C$ for lagoon sediments (Class 1) under brackish conditions after one year of anaerobic incubation. Unlike the Reindeer Island lagoon system, the lagoons on the Bykovsky Peninsula are located in the Yedoma domain (Jenrich et al., 2021; Strauss et al., 2021). Geographical differences and deposition mechanisms likely influence the colonisation of microorganisms, which in turn could explain the variations in $CO_2$ production.

As the openness of the lagoon increases, signifying a longer exposure of the lake basin to marine conditions, methane production decreases drastically. In our Class 5 lagoons only $CO_2$ production was detected, which was similar to or even higher than that in LAG13. This indicates a shift in dominant biogeochemical processes with prolonged marine influence, evidenced by the enrichment in $\delta^{13}C$ values (from -26.64‰ to -26.34‰) and a decrease in TOC content (from 6.38 wt% to 2.65 wt%). The porewater salinity of the studied lagoons exhibited considerable variation (Tab. 2) despite all having brackish surface

water salinity during the summer (Tab. 1). This variation was also highlighted by Solomon et al. (2000) and can be attributed to differences in pore size and water depth among the lagoons. For example, LAG3, which is shallower (1.3 m depth) and has sandy sediment, shows a lower pore water salinity (16.44 mS/cm). In contrast, the neighbouring LAG4, with a greater depth (4.6 m) and finer-grained sediment, exhibits a much higher pore water salinity (48.1 mS/cm). LAG3 tends to freeze to the bottom in winter due to its shallow depth, significantly impacting the salinity dynamics. During the formation of lagoon ice,

high saline brines form at the lagoon bottom (Angelopoulos et al., 2020; Jenrich et al., 2021). These brines can infiltrate into the sediment, as high pore water salinities observed in LAG4 suggest. Bottom-fast ice can freeze the underlying sediment, potentially expelling salts further down, resulting in maximum salinities at greater depths (Jenrich et al., 2021). In LAG4, which contains fine-grained sediments, the downward diffusion of salt is slower compared to lagoons with coarser-grained sediments, leading to higher salinities in the surface layers. The combination of these factors—ice formation, brine infiltration,

sediment grain size, and water depth—creates complex salinity profiles within these thermokarst lagoons, influencing biogeochemical and microbial processes.

Greenhouse gas production varies between lagoons within the same class. LAG4 (Class 3), under near-natural (marine) conditions, produces more than twice as much $CO_2$ as the brackish LAG14 (Class 3) (Fig. 4). However, when subjected to the same treatment, $CO_2$ production in both lagoons behaves similarly. The two lagoons differ in pore water salinity (31.5 vs. 18

450   g/L) and depth (4.6 m vs. 6.4 m), but they share similar biogeochemical and sedimentological parameters. Since the biogeochemical parameters in both lagoons are similar and the great water depth prevents bedfast ice formation, it is most likely that the composition of the microbial community is the reason for this difference, but no data is available to support this hypothesis. Despite their close proximity, Lagoons 3 and 16 differ in many parameters, which may cause the variation in GHG production. These differences include depth (1.1 m bedfast ice vs. 2.9 m floating ice), TOC (1.5% vs. 3%), C/N ratio (8.5 vs.





16.6), salinity (9.7 vs. 23.2 g/L), and DOC (7.8 vs. 26.7 mg/L). Under near-natural (brackish) conditions, Lagoon 3 produces 2.5 times more $CO_2$ than Lagoon 16, but no $CH_4$ production was detected within the experiment time frame. Many incubation studies have shown that methane production has a long lag time and can start much later than $CO_2$ production (Jenrich et al., 2024a; Jongejans et al., 2021; Knoblauch et al., 2013, 2018; Knorr and Blodau, 2009; Rivkina et al., 2007; Roy Chowdhury et al., 2015). Lagoon 3 is very shallow, and bedfast ice formation likely causes the upper sediment layers and the microbes within them to freeze. Freezing is a significant disturbance for microbes (Holm et al., 2020), especially for the slower-recovering and smaller methanogenic community. This is a possible reason why methane is produced in Lagoon 16 but not in the shallow Lagoon 3.

The observed differences in greenhouse gas production between lagoons within the same class highlight the potential influence of factors such as microbial community composition, salinity, and depth, while connectivity to the sea plays a crucial role in shaping the broader patterns of GHG production across different lagoon classes. Class 2 lagoons with low connectivity to the sea show the most dramatic difference between brackish and marine conditions, with significantly higher GHG production under brackish conditions, indicating that microbes are not yet adapted to higher salinities. Class 3 and 4 lagoons show a similar pattern, with a moderate GHG production under brackish conditions and a marked increase under marine conditions, indicating that microbes are already adapted to higher salinities at that stage of land-sea-transition. Very open, Class 5 lagoons maintain relatively high GHG production under both brackish and marine conditions, with slightly higher production under marine conditions. Lagoons of this class exhibit the highest GHG production stability across both treatments, implying that microbes in very highly connected lagoons are less sensitive to changes in salinity compared to less open lagoons.





## 4.2 CO₂ and CH₄ production along a land-sea transition gradient

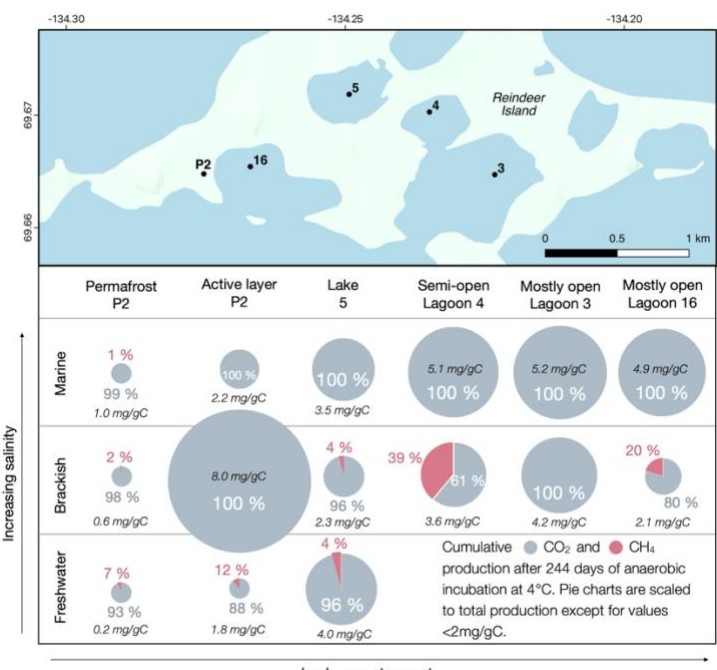

land - sea - transect

**Figure 5. GHG production along a land - sea - transect. Cumulative CO$_2$ and CH$_4$ production after 244 days of anaerobic incubation under fresh, brackish and marine conditions for the terrestrial sites and under brackish and marine conditions for the lagoons. Total production normalised to gC. Map: basemap Esri.**

The production of CO$_2$ and CH$_4$ varies significantly with increasing seawater (and thus salinity) influence along a land-sea transition gradient. Cumulative CO$_2$ and CH$_4$ production was measured after 244 days of anaerobic incubation under fresh, brackish, and marine conditions for terrestrial samples, and under brackish and marine conditions for the lagoons.

By incubating terrestrial permafrost, active layer, and lake sediment with freshwater, we simulated different stages of lake evolution reaching from a freshly formed pond after the first time of permafrost thaw to a young lake and an old lake respectively. The results demonstrated that the total GHG production increases as lakes evolve through these stages (Fig. 5). CH$_4$ production in terrestrial samples is highest under freshwater conditions, where methanogens are best adapted (Wen et al., 2017), especially in the organic-rich active layer (Fig. 3). In the active layer, the organic matter is less degraded than in the lake sediment. Further the sediment has thawed over many summers, so the microbial community is better adapted to current conditions than in the first time thawed permafrost, which shows the lowest cumulative GHG production across all treatments. This suggests that microbial communities in these sediments need more time to establish than was allowed in this experiment.The long-term incubation study by Knoblauch et al. (2013) has revealed that the labile organic matter can be mineralized within the first 3 months but the turnover of stable carbon pools may take several thousands of years.

The incubation experiment revealed that methane production is highest in LAG4, the most closed along the transect, under brackish conditions (Fig. 3). Compared to CO$_2$ production, CH$_4$ production starts after a long lag phase, assuming that sulfate



concentrations were too high in the beginning of the experiment. In the end of the incubation sulfate was depleted in LAG4, conditions which promote methanogene activity and therefore $CH_4$ production (Dar et al., 2008). As marine conditions intensified, $CO_2$ production continued to rise, whereas $CH_4$ production stopped under marine conditions. The higher availability of sulfate as an electron acceptor is promoting SRB over methanogens under marine conditions (Kristjansson and Schönheit, 1983).

The observed increase in $CO_2$ production under marine conditions contrasts with findings from an incubation experiment using sediment from lagoons along the Bykovsky Peninsula coast in NE Siberia (Jenrich et al., 2024a), where higher salinities led to a decrease in $CO_2$ production. They suggested that microbes were better adapted to brackish, near-natural conditions than to marine conditions. Unlike the lagoons in this study, which are part of an interconnected system, the Bykovsky Peninsula lagoons are more isolated. It is possible that marine microorganisms are better distributed in a lagoon system, allowing them to colonise upper soil layers and adapt more effectively to marine conditions than in isolated lagoons. However, microbial analyses are needed to confirm this hypothesis. Further, the results of the PCA indicate that  local conditions have the greatest impact on GHG production, therefore more studies are needed to get an understanding of the underlying mechanisms.

At Reindeer Island $CO_2$ production increased dramatically when the terrestrial active layer was inundated with brackish water for the first time, overshooting $CO_2$ production in lagoon sediments. This underlines that $CO_2$ production is highest in young, freshly formed lagoons or recently flooded coastal lowlands. In comparison, $CO_2$ production is much lower for active layer sediment incubated under marine conditions, showing that an extreme shift in salinity in a short timescale is not beneficial for microbial activity. The increase in $CO_2$ production from brackish to marine conditions in lagoon sediments demonstrates that once the former terrestrial microbial community has adapted to marine conditions during the slow process of seawater inundation, $CO_2$ production can pick up.

## 4.3 Incubation experiments in the context of Arctic coastal carbon dynamics

Placing the greenhouse gas production rates we measured into the larger context of Arctic coastal carbon dynamics is challenging due to the limited number of comparable studies. Tanski et al. (2019) used a similar incubation setup to investigate carbon dioxide production from eroding coastal permafrost. Their $CO_2$ production values were higher, however, their setting is not directly comparable since they incubated under aerobic conditions where carbon turnover is known to be faster. Similar to our study, Tanski et al. (2019) found that $CO_2$ production is higher with seawater than without and concluded that $CO_2$ production is promoted along the coastal zones. Our investigation has shown that this effect occurs not only under aerobic conditions at the coast but also under anaerobic conditions in the sediment below the water column. The availability of sulfate as an alternative electron acceptor promotes SRB and leads to a shift from a balanced $CH_4$ and $CO_2$ production in young, less connected lagoons towards purely $CO_2$ production with ongoing marine impact.

Contrary to our findings, Lougheed et al. (2020) report higher in situ $CO_2$ concentrations in the water columns of freshwater systems (thermokarst ponds, lakes, and rivers) compared to saline and brackish systems (ocean, coastal lagoon, and brackish rivers) on the Arctic Coastal Plain in Alaska, USA. However, they did not survey thermokarst lagoons, which are distinct from



coastal lagoons like the studied Elson Lagoon. Thermokarst lagoons originate from thermokarst lakes and therefore have a higher terrestrial signature than coastal lagoons. Elevated $CO_2$ concentrations in the nearshore waters of Elson Lagoon where the coastline is eroding highlights the role of terrestrial carbon and nutrient input (Tweedie et al., 2012). By studying the interface of sediment and the water column in Elson Lagoon, (Dunton et al., 2023) found a vertical gradient in $CO_2$

concentrations with high values at the lagoon bottom and a seasonal shift from net heterotrophic $CO_2$ release from sediments under ice to net autotrophic $CO_2$ uptake during break-up and open water. These findings underscore the complexity of carbon dynamics in Arctic coastal systems. So far we only know how much $CO_2$ and $CH_4$ is produced in thermokarst lagoon sediments under laboratory conditions. In-situ $CO_2$ concentration measurements of the water column would be needed to understand the carbon flux from the sediment to the atmosphere to predict their climate impact.

A total of 520 thermokarst lagoons exist along the coasts of the Laptev, East Siberian, Chukchi, and Beaufort shelf seas, including individual lagoons that are part of a larger lagoon system (Jenrich et al., 2024b). They are thus more than twice as abundant as the 216 identified coastal lagoons in the same region (Angelopoulos et al., 2021). More than half of the identified thermokarst lagoons are young, low connected lagoons (class 1 and 2) (Jenrich et al., 2024b). This study, along with the previous study by(Jenrich et al., 2024a), shows that these low connected lagoons, which are at the first stages of lake-to-sea

transition, have the highest GHG production and therefore should be the focus of further research.

Laboratory incubations offer a controlled environment that provides valuable insights into how GHG production is influenced by specific factors, such as salinity. This approach allows us to isolate and understand the impact of single environmental parameters. However, while laboratory settings are excellent for focused investigations, they do not replicate the full complexity of natural environments, which involve a wide range of factors and interactions. In natural settings, various

parameters, such as the duration of the open-water season, annual fluctuations in pH and water temperature, water depth, bathymetry, sediment and nutrient input from sources like the Mackenzie River, and plant-soil interactions, significantly affect GHG pathways in land-sea transitional areas. Due to the controlled nature of laboratory incubations, our study could not account for these additional variables. Despite this limitation, incubation experiments remain a valuable tool for studying GHG production and carbon dynamics in remote Arctic coastal environments. Future research should integrate laboratory

experiments with field studies to encompass seasonal variations, water depth, and sediment interactions. This combined approach will provide a more comprehensive understanding of carbon fluxes and enhance predictions regarding how Arctic coastal systems will respond to environmental changes.

**5 Conclusion**

We found significant variations in GHG production along a gradient from terrestrial to marine settings in Arctic coastal regions.

$CH_4$ production was high in terrestrial sediments under natural freshwater conditions due to methanogen adaptation. When Arctic permafrost lowlands are inundated with brackish water, $CO_2$ production accelerates. Young, less connected thermokarst lagoons, which comprise over 50% of the mapped pan-Arctic thermokarst lagoons, exhibited the highest $CH_4$ and $CO_2$



production. Their $CO_2$ equivalents can be up to 18 times higher than the open lagoons, attributed to high terrestrial organic carbon content and distinct microbial communities. As lagoons become more open, $CH_4$ production decreases or ceases, while
$CO_2$ production rises with increased seawater influence. $CO_2$ production was significantly higher under marine conditions, whereas $CH_4$ production was significantly higher under brackish conditions.

Moreover, we found that pore water salinity in lagoon sediments was highly variable, but not directly related to the openness of the lagoon. Instead, water depth and the formation of bedfast ice, along with the associated brine exclusion, had the greatest impact on pore water salinity and thus likely on the lagoon GHG production potential.

In the context of climate change, thermokarst lagoons forming along Arctic permafrost coastlines with ice-rich permafrost and abundant thermokarst lakes and basins are a part of the overall permafrost-climate-feedback loop. Arctic warming, permafrost thaw, and accelerating permafrost coastal erosion likely will increase thermokarst lagoon formation, amplifying carbon release and contributing to further warming. Sea level rise further accelerates the transition from terrestrial to coastal environments in permafrost coastal lowlands, enhancing $CO_2$ production. Understanding these processes is crucial for predicting and mitigating
the impacts of Arctic changes on global climate as well as capturing the full permafrost carbon picture in models.

**Data availability**

All data generated in this study will be available on the PANGAEA data repository. Data is currently in submission.

**Acknowledgements**

We thank G. Tanski for helping organise field trip logistics. We further thank Jimmy Kalinek and Nya for boating us and for
keeping us safe during fieldwork and remote camp life. We greatly thank our fieldwork assistant A. Flamand from Université du Québec à Rimouski for always having a helping hand and a bright mind. Thanks to A. Robertson for bringing a nitrogen bottle by helicopter which ensured keeping samples anoxic. For their essential help in the laboratories, we thank the Alfred-Wegener Institute Helmholtz Centre for Polar and Marine Research (AWI) and Helmholtz-Zentrum Potsdam Deutsches GeoForschungsZentrum (GFZ) lab technicians (J. Lindemann, A. Eulenburg, J. Serau, O. Burckhardt, S. Okolski).

**Funding**

MJ is funded by the DBU (project "Characterization of Organic Carbon and Assessment of Greenhouse Gas Emissions in a Warming Arctic") and the idea of this study is also a connected to the completed Changing Arctic Carbon cycle in the Coastal Ocean Near-shore (CACOON) project (a project focussing on the dynamic interface between land and ocean in the Arctic funded by the German Federal Ministry of Education and Research (BMBF; #03F0806A). The Alfred Wegener Institut
provided the baseline funding for this project and funded the Arctic fieldwork, as well as open access publication.



**Author Contribution**

MJ and JS designed this study. MJ and JS developed the overall coring plans for the Reindeer Island field campaign. MJ and JS conducted the field work. DW enabled field work due to his logistical support. MJ and FG performed laboratory analyses. SL supported the incubation experiments by supplying GC facilities. JW conducted statistical analyses. MJ led the writing of 590 the first draft of the manuscript. All co-authors contributed within their specific expertise to data interpretation as well as manuscript writing.

**Competing interests**

Susanne Liebner is a member of the editorial board of Biogeosciences (BG).

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
