# Peer review of "Rising Arctic Seas and Thawing Permafrost: Uncovering the Carbon Cycle Impact in a Thermokarst Lagoon System in the outer Mackenzie Delta, Canada"

_EGUsphere, 2024_

## Author Response (AR1)

**Response to reviewer 1**

**The following document answers each r*eviewer comment (RC)* with our author's response (AR)**

*RC: General comments: The authors studied greenhouse production along the land-sea transect with increasing salinity to explore how future sea level rise in the Arctic may alter the carbon cycling of permafrost region**. I found this study well-conceived, with interesting and relevant questions.** The **experiment** is also **well executed and seems to generate some very robust results**.*

AR: Thank you very much for your review and the positive evaluation.

*RC: Thus, I do not have any questions regarding the design or results of the study, but I do find the writing is a little hard to understand with convoluted sentences and repetitive descriptions, especially Discussion. The Discussion of this work is unnecessarily long, with large part of it not to interpret the results but repeating the description of the results. I suggest that the authors **largely shorten the Discussion** (ideally to 2/3 or 3/5 of current length), and **cite some recent studies to highlight the most interesting findings** of this study, and **emphasize the significance of this findings to future climate change**.*

AR: Thank you very much. We agree that there is potential to shorten the manuscript and to avoid convoluted sentences. As suggested, we have shortened the manuscript, particularly the discussion section. The revisions can be seen in the forthcoming revised version and the version with tracked changes.

We emphasize the significance of our findings in the discussion and conclusion as these examples show:

"In terms of CO2 equivalents, calculated based on 36 gCO2eq/gCH4 for a Global Warming Potential (GWP)100 (Balcombe et al., 2018), the CO2-eq of LAG13 is significantly higher—up to 18 times—compared to the open lagoons (e.g. LAG12). This is in line with the results of Jenrich et al. (2024a), where the highest CO2 and CH4 production was also detected for the most closed lagoon (Class 1 after Jenrich et al., 2024a) under brackish conditions. Consequently, young lagoons with high OM content and quality exert the greatest climate impact among the studied lagoons (Fig. S2)."

"More than half of the identified thermokarst lagoons are young, low connected lagoons (Class 1 and 2) (Jenrich et al., 2024b). This study, along with the previous study by Jenrich et al. (2024a), shows that these low connected lagoons, which are at the first stages of lake-to-sea transition, have the highest GHG production and therefore should be the focus of further research"

"In the context of climate change, thermokarst lagoons forming along Arctic permafrost coastlines with ice-rich permafrost and abundant thermokarst lakes and basins are a part of the overall permafrost-climate-feedback loop. Arctic warming, permafrost thaw, and accelerating permafrost coastal erosion likely will result in higher numbers of young, high emitting thermokarst lagoons, amplifying carbon release and contributing to further

warming. Sea level rise further accelerates the transition from terrestrial to coastal environments in permafrost coastal lowlands, enhancing CO2production." (Conclusion)

Since GHG production in thermokarst lagoons is a relatively new field of study, there are very few studies available to cite.

*RC: Thanks for the good work and interesting findings.*

AR: You are very welcome, and thank you for your helpful comments and time reviewing our manuscript

*RC: Specific comments:*

*1. RC: When presenting statistical analysis, it is good to **show the F values and sample size**, such that the audience can better evaluate the robustness of the results.*

AR: Thank you for spotting this. Within the text, we did indeed only report n in the methods section. In addition, we plotted a paired plot indicating each individual sample on top of the boxplots in Figure 2, and reported n in Figure 3c. We will make sure to report n wherever we present statistical analysis results in the revised manuscript text as well. We did not report results of any F-tests, such as those related to ANOVA. F-Tests also have quite strong assumptions, which are perhaps not met by our data. We can report V for the Wilcoxon test if required.

*2. RC: Map of the study sites was (the Reindeer Island and the surrounding lagoon system) was presented twice (first in figure 1c, and again in figure 4 left). It seems unconventional and really unnecessary. I wonder why the authors need to present it twice. Given the limited space for figures in the journal, I suggest to **replace one of the study site figures with the PCA results**, which is currently enclosed in the supplementary document but seems important to the study.*

AR: Thank you. According to your comment, we deleted the repetitive map in Figure 4.

*3. RC: I noticed that the Wilcoxon signed-rank test was used for paired samples for statistic comparisons, **why not paired Student's t-test**? Wilcoxon test requires the strong assumption of symmetric distribution of the data, I am curious if this assumption is met for the analysis?*

AR: Thank you for the comment. We conducted both paired Student's t-tests and Wilcoxon signed-rank tests. Both have data assumptions, and both yielded similar results. We do have a small sample size. For the paired Student's t-test, the data should approximately be normally distributed, and the test is sensitive to outliers. Wilcoxon's signed-rank test does not have this limitation, but usually requires larger sample sizes. Regarding the symmetry requirement, we thought that is basically the null hypothesis, with differences within each pair being symmetric around 0? This is not the case for our data, because the null hypothesis is not met. But it seems we might have misunderstood something? In the end, we decided to report the results of the Wilcoxon test, as that one is non-parametric and both tests gave the same results. In the revised manuscript, we add the sentence "While the t-test assumes approximate normality and is sensitive to outliers, the Wilcoxon test is non-parametric and

does not rely on these assumptions. Since both tests yielded the same results, we reported the Wilcoxon test outcomes." for clarity.

*4. RC: All analyses were performed in R, **please show the specific R packages** associated with each statistical analysis such as to increase the reproducibility of their work.*

AR: Thank you for the comment. We reviewed the methods section where we explained the statistical analyses and confirmed that we cited base R and the one additional package we used (ggplot2). It may have been unclear because we employed relatively simple statistical methods appropriate for our small sample sizes, relying on functions available in base R. We noticed that we did not explicitly name the function used for the Wilcoxon test (also from base R, with no additional package required), which we will address in the revised text.

*RC: Technical corrections: Please **pay attention to the format of terminology** (e.g. CH4, number subscripted, not CH4), and also make sure the presentation of **units or reference of figures/tables is consistent** throughout the manuscript (e.g. figure/figures, or fig./figs.).*

AR: In the revised version, all formats are checked and adjusted, as well as the consistency of units and reference figures etc. Thank you again for the careful review and the time you spent helping us to improve our manuscript.
* * *
**Response to reviewer 2**

**The following document answers each r*eviewer comment (RC)* with our author's response (AR)**

*RC: Jenrich and colleagues have sampled from a set of changing permafrost affected lagoons and small-delta-island lakes in Canada's NW Arctic and conducted a long-term, low-temperature incubation study exploring carbon mineralization (and net methane and carbon dioxide production) and potential controls related to the different land/marine-forms. I admittedly do not know the geomorphological context of the study well at all, but have experience in controls on microbial biogeochemistry in terrestrial wetlands, including permafrost affected sites. Overall, I found the manuscript and study interesting, seemingly very relevant (again with the caveat that I don't know arctic marine coastal/delta island geomorphology), and easy to read.*

AR: Thank you very much for your evaluation and time spending on our manuscript. We appreciate your helpful comments. Please see out point-by-point replies in the following:

*RC: The manuscript at large is long and some parts wordy based on my preference, but generally (with a few small objective formatting errors and typos), there isn't much technically incorrect writing, and I would defer to the authors and journal requirements about the general nature of the written presentation.*

AR: Thank you very much. You are in close agreement with the other reviewer, and we are happy to say that we have shortened the manuscript carefully, with a special focus on the

discussion. Please check the revised version as well as the version with tracked changes as a proof for that.

*RC: Some of my specific technical suggestions are:*
*For a more lay reader to the geomorphological context like myself, consider expanding the explanation of the historical and contemporary transition of isolated lakes to thermokarst and increasingly marine-connected lagoons.*

AR: Thank you. We added "Thawing of the ice-rich permafrost led to water-filled thaw depressions or thermokarst lake formations that can extend over 20 m deep. Marine transgression during the Holocene period (last 10,000 years) flooded the region creating submerged basins. In addition, lake expansion, sea level rise, and coastal erosion caused the inundation and eventual connection of individual lakes to each other and ocean, forming the current lagoons system that separates Reindeer Island from Richards Island." (section 2, paragraph 2) to the revised version.

*RC: I generally really like your figures, but I did struggle to see the 1972 coastline as highlighted in the Figure 1 legend. I suppose I was expecting to see what was a clearly a lake then that is now 52y later a lagoon, but did not.*

AR: You are correct—the coastline changes were difficult to observe within the lagoon system. Coastal erosion was most pronounced along the open sea coasts, where changes were more evident. In contrast, the slower coastline change rates within the lagoon system indicate that lake-to-lagoon transformations take longer than 50 years. To avoid confusion, we chose to omit the 1972 coastline data.

*RC: Some additional context about the timing of the coastal erosion and permafrost change in the study sites I think would be useful.*

AR: Sure, we are happy to have this included in the added sentence mentioned above (see section 2, paragraph 2 as mentioned in the previous comment).

*RC: There was a prior expectation in the introduction about the role of sulfate reduction (more prominent in S rich marine sediments) influencing methanogenesis that didn't fully play out. There could be a disconnect between a single sulfate measurement in bulk cores at that beginning of the incubation that isn't relevant in incubations running for multiple hundreds of days in closed anoxic systems. Overall, it did seem that the stronger the influence from the marine environment the lower the methane (and increasingly important carbon dioxide as a mineral anaerobic microbial product).*

AR: Thank you for this statement. Yes, this is indeed what we see as stated in line 430 of the submitted manuscript. To make this more clear we rephrased the first sentence of section 5.1 as following: "We found that the production of greenhouse gases shifts from a balanced CH4:CO2 ratio to the exclusive production of CO2 as the lagoons become increasingly connected to the sea. Thus, as expected, an increasing influence of sulfate reduction seems to inhibit methanogenesis."

*RC: In other marine sediments, anaerobic respiration pathways have been shown to be stratified by depth with electron acceptor recharge occurring from the top down because of*

*the oxygenated, mineral-rich water column. SRB might occupy a small horizon, and bacterial fermentations and methanogenesis occur in a slightly deeper horizon (on the scale of millimeters in low turbulent/laminar flowing waters). Are there related laboratory artifacts due to the static/sealed small jars used (v say a column approach with sediment below water and a source of electron acceptor recharge)? I think this warrants some explanation and discussion.*

AR: Thank you for raising this point. Of course, lab incubations are another potential source of artificial artifacts. This risk actually starts with the sampling on site itself, taking the sample out of the natural open system. Following your suggestion, we added the following paragraph to the method section 3.4 "To assess anaerobic respiration pathways in a controlled environment, we used sealed glass jars with 10g sediments. This setup allowed us to maintain anoxic conditions typical of marine sediment environments. However, the static, sealed nature of these jars may limit natural electron acceptor recharge observed in stratified sediments under open systems with dynamic hydrology. This limitation is noted as a potential artifact when extrapolating findings to natural, vertically stratified marine sediments, but gives the advantage of having a controlled setting in the lab to decipher the changes under controlled conditions, e.g. salinity gradients" as well as to the discussion by stating "In situ, vertical stratification of redox potential layers is influenced by dynamic recharge of electron acceptors from overlying oxygenated and mineral-rich water columns, which cannot be replicated in sealed jars." (last paragraph in section 5.3).

These processes are a research topic itself and we appreciate contributing to future studies employing column setups with a continuous supply of electron acceptors, which could provide complementary insights into the effects of hydrodynamic conditions on microbial activity. We further added: "Future research should integrate laboratory experiments with field studies to encompass seasonal variations, water depth, and sediment interactions, and explore column-based approaches with continuous electron acceptor recharge to better replicate natural stratification."

*RC: Following from 3 above, are there other anaerobic respiration pathways (that would influence proportions of carbon dioxide v methane and methanogenesis overall) beyond sulfate reduction that deserve some attention?*

AR: Thank you for raising this important point. Of course there are a variety of processes. As you and the other referee posted that the discussion is detailed already, we try to add this comment as short as possible: "Beyond sulfate reduction, other anaerobic respiration pathways may influence the balance of $CO_2$ and $CH_4$ production in sediments. Iron reduction, driven by Fe(III)-reducing microbes, can outcompete methanogenesis in Fe(III)-rich environments, favoring $CO_2$ production. Similarly, nitrate reduction may suppress methanogenesis through substrate competition, though nitrate levels in permafrost are typically low. Methanogenesis itself occurs via different pathways: hydrogenotrophic methanogenesis, which depends on $H_2$ availability and dominates in nutrient-poor conditions, and acetoclastic methanogenesis, reliant on acetate—a substrate that can accumulate in thawing permafrost. Additionally, anaerobic oxidation of methane (AOM), a process that recycles $CH_4$ into $CO_2$ using electron acceptors like sulfate or iron, may play a role, as observed in similar Arctic lagoon systems. However, due to the lack of measurements for Fe(III), nitrate, acetate, and methane fluxes in this study, these pathways

remain speculative and warrant further investigation." This is now included in section 5.2 of the revised version.

*RC: In the ecosystems in-situ, and as represented in-vitro in your incubations, is there an expected or **potential role for anaerobic methane oxidation**? There is a large contrast of the low net methane measured in the marine influenced samples: does this imply low activity of methanogens per se, that produced methane is being consumed, or both? Does it matter?*

AR: Yes, the major reason seems to be the decrease in methanogen activity: Previous studies have shown that the abundance of methanogens decreases significantly with increasing saltwater influence (see Yang et al., 2023 for more detail on AOM in lagoon sediments); therefore, it is expected that less methane is produced.

*RC: I would like to see some more overall speculation in the Discussion what patterns seen in the lab-based, long-term, static jar incubation approach might mean in the field (both how the lab system does and does not represent field conditions and resources). E.g., considering redox potential layering in situ, the fate of mineral C products in overlying alkaline, oxygenated water, etc.*

AR: Thank you very much for your very helpful review. Balancing the request to shorten especially the discussion while making full use of our unique dataset, we have aimed to include the additions above in a way that aligns with your suggestions and enhances the discussion meaningfully. We hope these revisions meet your expectations.

At the end of the discussion we added this new paragraph on future pathways might help for clarity: "The patterns observed in this long-term incubation study provide insights into anaerobic respiration and carbon dynamics but also highlight some limitations in representing field conditions. In situ, vertical stratification of redox potential layers is influenced by dynamic recharge of electron acceptors from overlying oxygenated and mineral-rich water columns, which cannot be replicated in sealed jars. Pathways such as iron and nitrate reduction, hydrogenotrophic and acetoclastic methanogenesis, and anaerobic oxidation of methane (AOM) play distinct roles in field environments, where their activity is modulated by substrate availability and redox gradients. In the jars, the static environment may limit competition among pathways, potentially amplifying methanogenesis relative to iron or nitrate reduction. While hydrogenotrophic methanogenesis likely dominates in nutrient-poor conditions of Arctic sediments, the fate of mineralized carbon products (e.g., $CO_2$) in overlying alkaline, oxygenated waters—where precipitation of carbonates can occur—is not captured in this setup. Additionally, AOM, observed in comparable systems like Bykovsky lagoons (Yang et al., 2023), might reduce net $CH_4$ emissions in situ but remains speculative without methane flux data. These limitations underscore the need for complementary field studies using dynamic systems, such as sediment columns with continuous electron acceptor recharge, to validate lab-based findings and better understand how they translate to natural sedimentary environments."